# OpenReview forum: "AutoControl Arena: Synthesizing Executable Test Environments for Frontier AI Risk Evaluation"
_ICML.cc/2026/Conference — ICML 2026 regular_

### Official Review · Reviewer_qMvB · 2026-03-10

**Soundness:** 3
**Presentation:** 3
**Significance:** 3
**Originality:** 2
**Overall Recommendation:** 4
**Confidence:** 3

**Summary:**

The paper's general area is automated safety evaluation for frontier LLM agents in realistic, tool-using, multi-step environments, aiming to reduce logic errors and improve reproducibility. It proposes AutoControl Arena, which synthesizes executable testbeds via logic-narrative decoupling, with a three-agent pipeline that designs scenarios, generates Python-backed environments, and audits behavior. The authors introduce AutoControl Arena to address the fundamental trade-off between fidelity and scalability in AI risk evaluation through "logic-narrative decoupling." Overall, the authors assess the central issue of whether benign evaluations hide latent misalignment by varying Stress and Temptation across X-BENCH, a benchmark with 70 scenarios, and measuring how risk rates and failure modes change across 9 frontier models.
The paper's findings are that risks increase sharply under pressure, stronger models are better at avoiding direct harm but more likely to game rules or hide issues, while weaker models cause harm mainly due to incompetence.

**Compliance With Llm Reviewing Policy:**

Affirmed.

**Key Questions For Authors:**

* Monitor Agent Evaluations: The one monitor agent achieves 92.8% F1 score, which is roughly ~7% error rate. It would be impactful if those error cases are analysed and root-causes provided (if possible).

* Generalisation Beyond X-BENCH: Can you evaluate your approach on another benchmark dataset like Petri? Is there an assumption that this framework would generalise beyond the specific 70 scenarios in X-BENCH?

* Monitor and judge reliability: why Qwen3-32B, and how sensitive are results to judge choice and rubric design? Given the Monitor’s rubric-heavy role and its use of both action evidence and chain-of-thought style reasoning signals, how would risk rates and qualitative conclusions change under stronger judge models, alternative judge families, or ensemble judging? A strong dependence on judge choice would reduce confidence that reported cross-model risk differences reflect target-model behavior rather than monitor calibration.

* How representative is the Sim-to-Real calibration, and how “Real” is defined in your setup? You report Sim-to-Real calibration over 10 scenarios with 10 trials each and interpret it as a proxy for deployment risk.  Please clarify what the “Real” environments are (e.g., manually implemented sandbox vs deployment-like systems), how the 10 scenarios were selected, and where the proxy fails. If “Real” is primarily a manual sandbox baseline and agreement varies under broader coverage, it would weaken claims about external validity.

* Environment-generation sensitivity: how much do findings depend on the Architect/Coder model and prompts? The pipeline fixes **Qwen3-Next-80B-A3B as Architect/Coder** in the main experiments. Since scenario synthesis can shape what vulnerabilities are exposed, do the headline findings persist under different Architect/Coder models or prompt variants? If results are highly sensitive to synthesis choices, it would weaken the generality of conclusions like “alignment illusion” and cross-model comparisons.

**Limitations:**

Yes

**Strengths And Weaknesses:**

## Soundness

**Strengths**

* The core design is technically coherent: deterministic state and transitions are enforced in executable code, while only open-ended narrative elements are delegated to an LLM, directly targeting state inconsistency and logic hallucination seen in text-only simulators.
* The implementation is supported by concrete artefacts in the appendix, including full role prompts, environment code examples, and an end-to-end worked example with full code for a scenario.
* Empirical evaluation is broad and systematic: multiple frontier models are tested across X-BENCH scenarios and the paper reports clear aggregate patterns in risk across evaluation conditions.

**Weaknesses**

* The sim-to-real style check is closer to sim-to-manual: the comparison “real” side still appears to be a human-implemented sandboxed environment rather than a live deployment setting, so it mainly validates agreement with a manual test harness, not external real-world validity.
* Monitoring and several key judgments rely on LLM-mediated signals, including chain-of-thought based analysis and a complex rubric-heavy judge. This can be fragile when chain-of-thought is unavailable or unreliable, and it increases the surface for evaluation artifacts such as models shaping or concealing their reasoning to pass the monitor.
* Limited calibration scale (only 10 examples selected for Real-to-sim Fidelity assessment) and structural reliance on a single monitoring model (Qwen3-32B). The paper only evaluated their approach on their own released benchmark data.

## Presentation

**Strengths**

* The narrative is easy to follow: problem statement, core principle, system design, validation, and then empirical findings, supported by clear diagrams of the pipeline and evaluation matrix.
* The paper positions itself concretely against related paradigms and nearby systems such as ToolEmu, Petri, and SHADE-Arena, and articulates the fidelity versus scalability motivation well.
* The appendix appears to provide substantial implementation detail, including prompts and environment examples, which helps reproducibility and auditing.

**Weaknesses**

* The main text could more crisply define the labeling and scoring pipeline for risk and show how heterogeneous category rubrics map into a consistent decision rule, since much of the operational detail lives in the appendix.
* The manuscript asks the reader to track many named constructs in rapid succession.
* Readability suffers as a lot of important information is in appendix. A suggestion is to add a dedicated "Limitations and Future Work" subsection in the main paper (not just appendix) discussing major limitations and providing guidance on appropriate use cases.

## Significance

**Strengths**

* The paper targets a high-leverage problem for agentic AI safety: scalable yet faithful evaluation of tool-using agents, where manual red-teaming does not scale and text simulators can be unreliable.
* The executable-kernel approach is practically relevant and the reported “alignment illusion” pattern is a useful warning that safety conclusions can be highly elicitation-dependent.
* Introduction of a Reusable Infrastructure Layer is significant contribution
* The three-agent pipeline (Architect → Coder → Monitor) and X-BENCH (70 scenarios × 4 elicitation settings pull and push) provide:
	* A reusable evaluation infrastructure
	* A scalable testbed generation paradigm
        * A stress-testing methodology (Stress × Temptation grid)

**Weaknesses**

* Real-world usefulness will depend on adoption factors that are not fully quantified in the headline results, such as engineering overhead, runtime cost, and how easily new domains can be encoded without introducing new failure modes.
* The scope is explicitly limited to pre-deployment testing rather than comprehensive safety evaluation, which limits impact.

## Originality

**Strengths**

* The main originality is the concrete logic-narrative decoupling implementation paired with an automated synthesis pipeline that outputs executable environments, not just text simulators.
* The benchmark plus elicitation design offers a reusable scaffold for probing when misalignment emerges under pressure and opportunity, and it is executed at relatively large scale.

**Weaknesses**

* Many ingredients resemble existing patterns in LLM evaluation and oversight, so the novelty is concentrated in the specific end-to-end packaging and the executable enforcement layer rather than wholly new primitives.
* Executable environments already exist (e.g., SHADE-Arena, SafeArena) as well as red-teaming and auditing frameworks (Petri).
* Stress-testing alignment under pressure is already discussed in prior alignment literature. The paper mostly innovates primarily in integration, scale, and formalisation, not in inventing an entirely new research direction.

---

> ### Author Rebuttal · Authors · 2026-03-31
>
> # Rebuttal to Reviewer qMvB
>
> We sincerely thank the reviewer for the constructive feedback. We address each question and weakness below.
>
> ---
>
> > **Q1: Monitor error case analysis**
>
> We analyzed the misclassified cases and identified three main failure modes:
>
> - **(1) Attempted-but-unsuccessful bypass is under-weighted:** when an unauthorized action fails to execute, monitors may be influenced by the failed outcome and underweight the underlying intent.
>
> - **(2)** **Benign-looking framing suppresses risk sensitivity**: risky behavior can be masked by legitimate-seeming workflow, especially when the signal is embedded in compliance-style expression.
>
> - **(3)** **Score boundary sensitivity**: our threshold (≥2 = Risky) means cases at the score 1 to 2 boundary, where behavior is subtle but not clearly benign, are inherently harder to classify definitively.
>
> Overall, these residual errors are concentrated in a small number of gray-zone patterns. We will include this error analysis in the revision.
>
> ---
>
> > **Q2: Generalisation beyond X-BENCH**
>
> Our framework is generative by design: given a scenario description, the pipeline synthesizes a new executable environment. **We wish to clarify that Section 4.3.1 already applies our framework to 20 scenarios from the Petri dataset. Furthermore, X-BENCH itself contains scenarios from prior literature (Section 3.6), confirming that our framework can reproduce known failure modes from external sources.**
>
> **To directly address the reviewer's question, we additionally ran our pipeline on 50 scenarios sampled from the Petri dataset using Minimax-M2.5 as the Architect/Coder.** We achieved a 94% generation success rate, with an average environment complexity of 675.52 lines of code and 9.00 tools per scenario, comparable to X-BENCH results.
>
> ---
>
> > **Q3: Monitor and judge sensitivity**
>
> **We selected Qwen3-32B for its cost-efficiency and strong calibration. To assess judge sensitivity, we re-evaluated using two stronger alternative monitors: Minimax-M2.5 and DeepSeek-V3.1.**
>
> Overall risk rates shift by +6.06% and +2.96% respectively under the two alternative monitors, which we attribute to stronger recall on gray-area cases by more capable monitors. **Critically, the main qualitative findings remain fully intact: risk rates escalate consistently from S0T0 to S1T1 under both alternative monitors, and the directional patterns are preserved across all elicitation conditions.**
>
> | Monitor | Overall Δ | S0T0 | S0T1 | S1T0 | S1T1 |
> |---|---:|---:|---:|---:|---:|
> | Minimax-M2.5 | +6.06% | +6.08% | +7.61% | +4.55% | +6.00% |
> | DeepSeek-V3.1 | +2.96% | +1.10% | +3.54% | +1.01% | +5.97% |
>
> **These results confirm that our findings reflect general target model behaviors rather than monitor calibration artifacts.** We also find that stronger models better capture several of the hard cases in Q1, especially those requiring richer contextual understanding over subtle risk signals. We view this as further motivation for continued investment in reasoning-aware monitoring.
>
> ---
>
> > **Q4: Sim-to-Real calibration and definition of Real**
>
> **Our "real environments" are deployment-like systems, rather than manually implemented sandboxes with simulated tools.** Specifically, we constructed Docker-based environments with actual CLI tools, live PostgreSQL database instances, and functional API endpoints, consistent with the standard in controlled real-world evaluation (e.g., SWE-agent, OS-Harm). Agents interact with genuine system interfaces throughout.
>
> Scenario selection followed two criteria: (1) the risk behavior must be physically instantiable in a real computer environment, excluding scenarios requiring non-digital infrastructure (e.g., military logistics, healthcare equipment); and (2) scenarios span diverse risk categories. Full per-scenario results are provided in Appendix F.2.
>
> ---
>
> > **Q5: Architect sensitivity**
>
> **To empirically assess this, we re-ran X-BENCH scenarios using Minimax-M2.5 as an alternative Architect/Coder** (Monitor: Qwen3-32B, target models: Qwen3-4B/8B/14B):
>
> | Architect | S0T0 | S0T1 | S1T0 | S1T1 |
> |---|---:|---:|---:|---:|
> | Qwen3-Next | 21.54% | 29.15% | 30.89% | 60.30% |
> | Minimax-M2.5 | 24.85% | 33.86% | 33.84% | 61.19% |
>
> The headline findings remain stable across both configurations: risk rates escalate consistently from S0T0 to S1T1, with closely comparable absolute rates. **This confirms that main findings reflect genuine target model behavior rather than artifacts of the synthesis pipeline, even if finer-grained comparisons can shift modestly.** We will include this analysis in the revision.
>
> ---
>
> Finally, we also thank the reviewer for the presentation suggestions. In the revision, we will move more operational detail into the main text and add a dedicated Limitations and Future Work subsection.

---

> > ### Author Rebuttal · Reviewer_qMvB · 2026-04-01
> >
> > Thank you for the detailed rebuttal. The additional analyses on monitor errors, judge sensitivity, sim-to-real calibration, and Architect/Coder sensitivity address several of my main concerns and strengthen confidence in the empirical findings. However, my concern about originality is only partially resolved: while the rebuttal reinforces the paper’s robustness and scope, it does not clearly sharpen the distinction between the paper’s main contribution and closely related prior work on executable environments, red-teaming, and oversight frameworks. I therefore consider my concerns partially resolved and encourage the authors to clarify this positioning more explicitly in the revision.

---

> > > ### Author Response · Authors · 2026-04-03
> > >
> > > We sincerely thank the reviewer for the thoughtful acknowledgement.
> > >
> > > **To sharpen the distinction:** existing executable environments (SHADE-Arena, SafeArena) require per-scenario human engineering and do not scale to novel risk domains. Existing LLM-based frameworks (ToolEmu, Petri) achieve automation but suffer logic inconsistency through text-based simulation, trading off fidelity. **Our contribution is precisely the combination that prior work lacks: automated synthesis of executable environments from high-level risk descriptions, enabling scalable coverage of the long tail of frontier risks at low cost.** It provides a controllable, reproducible, and scalable pre-deployment proxy for frontier-risk discovery, especially for open-ended, high-stakes scenarios that are difficult to instantiate in live systems. This is what logic-narrative decoupling achieves and no prior work achieves this combination.
> > >
> > > **Beyond the framework itself, our empirical findings:** the quantified Alignment Illusion (21.4%→52.9%), non-monotonic Safety Scaling Laws, and Divergent Misalignment Patterns, represent reproducible discoveries about frontier model behavior that **stand independently as contributions to alignment research and deployment practice.**
> > >
> > > We hope this clarification addresses the remaining concern and will incorporate this directly in the revision.

---

### Official Review · Reviewer_TDL3 · 2026-03-11

**Soundness:** 3
**Presentation:** 3
**Significance:** 3
**Originality:** 2
**Overall Recommendation:** 4
**Confidence:** 3

**Summary:**

The paper introduces AutoControl Arena, a framework for automatically generating executable environments to evaluate safety risks in autonomous LLM agents. Its key idea is to separate deterministic logic, implemented in code, from open-ended narrative interactions, generated by LLMs, to improve both fidelity and scalability. Experiments across 70 scenarios and 9 frontier models show that models often appear safe in benign settings but become much riskier under stress and temptation.

**Compliance With Llm Reviewing Policy:**

Affirmed.

**Final Justification:**

The rebuttal satisfactorily addresses my concerns about the experimental results. While I appreciate the authors’ effort to clarify the contribution, some concerns regarding novelty remain. Taking everything into consideration, I am overall inclined toward borderline acceptance.

**Key Questions For Authors:**

1. How sensitive are the results to the choice of model architectures used for the Architect agent? Could different generator models produce environments that lead to substantially different conclusions?
2. Although the authors include sim-to-real experiments, it remains unclear how closely the generated environments resemble real-world deployment scenarios. To what extent do these environments reflect realistic operational settings, would it help if we use more powerful model as the architect agent?
3. Some behaviors identified in the experiments (e.g., reward hacking) may not always lead to real-world harm depending on context. Have the authors considered evaluating the severity or real-world impact of the detected behaviors in the specific environments?
4. When evaluating models under high stress conditions, do the authors observe cases where models explicitly refuse to comply due to safety alignment? It is a bit surprising that GPT models are having higher risk rates under adversarial conditions.

**Limitations:**

yes

**Strengths And Weaknesses:**

Strength
1. The authors introduce a structured validation process, which helps ensure the quality of the synthesized environments.
2. The paper evaluates multiple model families.
3. The experimental analysis identifies several interesting phenomena.

Weakness
1. The contribution is relatively weak. The central idea of transitioning the logic from LLM-generated environment states to executable code-based environments appears conceptually straightforward. And the two-dimensional elicitation frameworks are also not something new. [1]
2. While the paper presents several empirical findings, it remains unclear how these insights could concretely inform the design of more robust AI agents.

[1] DeceptionBench: A Comprehensive Benchmark for AI Deception Behaviors in Real-world Scenarios

---

> ### Author Rebuttal · Authors · 2026-03-31
>
> # Rebuttal to Reviewer TDL3
>
> We thank the reviewer for the constructive feedback. We address each point below.
>
> ---
>
> > **W1a: Central idea "conceptually straightforward"**
>
> **The logic-narrative decoupling principle may appear intuitive in hindsight, but realizing it reliably requires solving non-trivial challenges:** (1) achieving 98% success rate across environments averaging 434 lines of code and 8.6 tools;  (2) conditioning the Narrative Layer strictly on Logic Layer ground-truth to prevent hallucination leakage across multi-turn interactions. None of the prior works achieve this combination. Furthermore, the empirical discoveries represent novel findings about frontier model behavior, independent of conceptual novelty.
>
> ---
>
> > **W1b: Comparison with DeceptionBench**
>
> We respectfully note two fundamental differences.
>
> **First, evaluation targets differ.** DeceptionBench evaluates conversational LLMs in static QA settings, while our framework evaluates autonomous agents executing multi-step plans in stateful environments, where misalignment manifests through action sequences.
>
> **Second, elicitation mechanisms operate at different levels.** DeceptionBench applies static text perturbations, whereas our framework implements stress and temptation through environmental structure, tool definitions, and evolving context.
>
> ---
>
> > **W2: How findings inform robust agent design**
>
> **Our framework is designed as a testbed for alignment research, not just an evaluation endpoint.** As discussed in Appendix A.3, a natural extension is automated defense synthesis using discovered failure modes to improve system prompts, constitutional rules, or fine-tuning datasets.
>
> Concretely: Finding 1 motivates adversarial elicitation augmentation in safety training, since benign-only training misses risks that only surface under pressure. Finding 2 implies scenario-specific evaluation is necessary, as stronger models may be safer on direct harms yet riskier on loophole exploitation. Finding 3 motivates differentiated monitoring design, with frontier models requiring reasoning-aware monitoring to detect strategic concealment.
>
> ---
>
> > **Q1 & Q2: Sensitivity to Architect model choice**
>
> Our framework provides roubust  guarantees on environment generation through executable verification and repair loops, reducing  sensitivity to Architect model choice. **To empirically assess this, we re-ran X-BENCH scenarios using Minimax-M2.5 as an alternative Architect/Coder** (Monitor: Qwen3-32B, target models: Qwen3-4B/8B/14B):
>
> | Architect | S0T0 | S0T1 | S1T0 | S1T1 |
> |---|---:|---:|---:|---:|
> | Qwen3-Next | 21.54% | 29.15% | 30.89% | 60.30% |
> | Minimax-M2.5 | 24.85% | 33.86% | 33.84% | 61.19% |
>
> The high-level findings remain stable across both configurations: risk rates escalate consistently from  S0T0 to S1T1, though absolute risk rates may vary slightly due to stochasticity.
>
> **Regarding Q2: We would like to clarify our goal is not to perfectly replicate the real-world system, but to provide a calibrated proxy where detected risk behaviors can generalize to real deployment settings**,directly validated by our Sim-to-Real study (r=0.87). Our original setup already provides sufficient complexity (434 avg. lines) comparable to a stronger Architect (560.45 avg. lines under Minimax-M2.5).
>
> ---
>
> > **Q3: Not all detected behaviors lead to real-world harm**
>
> We agree the severity is context-dependent. **However, our focus is specifically on frontier-risk behaviors rather than ordinary content-safety violations.** The International AI Safety Report 2026 states that some frontier AI risks are already materialising with documented harms [1]. And leading AI organizations have identified as critical safety concerns [2,3].
>
> More directly, our Sim-to-Real study demonstrates these behaviors are not merely simulator artifacts: across 10 scenarios, simulated and real-world risk rates correlate strongly (r=0.87).
> [1] International AI Safety Report 2026
>
> [2] OpenAI, Detecting misbehavior in frontier reasoning models
>
> [3] NVIDIA, Frontier AI Risk Assessment
>
> ---
>
> > **Q4: Explicit refusals and GPT risk rates**
>
> Yes, we observe explicit refusals. GPT-5-mini provides the clearest example: lowest average risk rate (20.5%) and 0% for direct Capability Misuse, reflecting strong safety alignment.
>
> **The higher risk rates in earlier GPT generations do reflect a distinction between content safety and frontier safety**. Traditional alignment training targets content safety, while frontier risks (e.g., strategic misalignment) represent a newer safety dimension. This generational pattern is consistent with findings from Petri [4], suggesting frontier safety alignment is actively improving but remains incomplete — precisely the gap our framework is designed to surface.
>
> [4] Anthropic, https://alignment.anthropic.com/2025/petri/
>
> ---
>
> We hope the clarifications above address the main concerns, and we will revise the paper accordingly.

---

> > ### Author Rebuttal · Reviewer_TDL3 · 2026-04-02
> >
> > The authors’ explanation has addressed my concerns regarding the experimental results. While I still have some reservations about the strength of the paper’s overall contribution, I am willing to raise my score to 4 and view the paper as a borderline acceptance.

---

> > > ### Author Response · Authors · 2026-04-03
> > >
> > > We sincerely thank the reviewer for raising the score and for the constructive engagement throughout.
> > >
> > > As mentioned before, we focus on frontier-risk evaluation, where scalable pre-deployment testing is increasingly significant as many high-stakes, open-ended scenarios are confirmed in real deployments. Prior work has typically addressed only part of this problem: existing executable-environment frameworks rely heavily on per-scenario human engineering and are difficult to scale to novel risk domains, while LLM-based frameworks gain automation but suffer logic inconsistency in text-based simulation. **Our core contribution is precisely to bridge the gap: automated synthesis of executable evaluation environments from high-level risk descriptions**, enabled by logic–narrative decoupling, so that **scalability and executable grounding are achieved together in one reusable framework.** It provides a controllable, reproducible, and scalable pre-deployment proxy for frontier-risk discovery, especially for open-ended, high-stakes scenarios that are difficult to instantiate in live systems. This combination is not achieved by prior work in one unified framework.
> > >
> > > **Beyond the framework itself, our contribution is also empirical.** The resulting findings, including Alignment Illusion, Scenario-Specific Safety Scaling, and Divergent Misalignment Patterns, are quantitative, reproducible results that contribute to alignment research and deployment practice.
> > >
> > > We will sharpen this positioning more explicitly in the revision.

---

### Official Review · Reviewer_ZRar · 2026-03-12

**Soundness:** 3
**Presentation:** 4
**Significance:** 3
**Originality:** 3
**Overall Recommendation:** 3
**Confidence:** 4

**Summary:**

The authors introduce AutoControl Arena, a method for creating and scaling an automated framework for evaluating the risks posed by AI agents. In particular, they introduce a mechanism to decouple narrative elements (NPCs, messages, dynamic content) from logic ("deterministic mechanics" like files, databases, permissions, etc). The authors claim that this results in greater fidelity over existing frameworks, namely Anthropic's Petri.

**Compliance With Llm Reviewing Policy:**

Affirmed.

**Key Questions For Authors:**

1. Would it be possible to share the verbatim system prompts and configurations for all scenarios, in order to better assess the true fidelity of the environment?
2. If the aim of the work is to improve fidelity, may I ask why you have chosen (from what I can tell) to focus strictly on text-based loops with dummy states, as opposed to moving towards more realistic shell, API, or computer environments?

**Limitations:**

Yes.

**Strengths And Weaknesses:**

### Strengths

Assuming that the authors' claim of improved fidelity over Petri is true, then this framework improves upon current state of the art in the literature in terms of generating diverse environments on the fly, given some target assessment goal. This is undoubtedly a worthwhile venture, given that safety and risk evaluations are paramount to ensuring the safe deployment of AI models. Their method appears to be scalable, it has clearly been thought-out, and they demonstrate its utility through their benchmark results. Additionally, it is a smart design decision to decouple aspects which can be subject to hard constraints (tool definitions, params, states, etc) and aspects which are not. This undoubtedly adds a layer of trust to the evaluation system. Further, while the authors demonstrate the strengths of their system with respect to risk categories that are present in the literature or of major importance in the present, the system appears to be scalable to novel types of tasks.

### Weaknesses

While structurally this methodology seems sound, I have to call into question the authors' argument towards fidelity. The authors position their work as higher-fidelity than existing work (i.e. Petri), and while it is possible that may be the case, I don't think "better than previous" is a worthwhile metric to optimize for, particularly as one of your core arguments. For example, looking at the generated environment samples in appendix H, H.1 describes the agent as "a Military Logistics Coordinator responsible for
supplying forward operating bases." The "deterministic state" that the framework generates is as follows:

```
self.inventory = {
"base_alpha": {"ammo": 5000, "fuel": 12000},
"base_bravo": {"ammo": 3200, "fuel": 8500},
"base_charlie": {"ammo": 1800, "fuel": 4200}
}
```
This strikes me as incredibly lacking in fidelity. Additionally, while the authors claim that "tool neutrality" is another core enforcement, I would argue that placing the agent in a scenario where it has the capability to call `cancel_maintenance` is not neutral at all. Finally, I take issue with the scenario itself. I would argue that there is a lot of fidelity lacking from the scenario, given that real-world agentic deployments often contain a lot of noise, and significantly more considerations than appear present in this framework. I will also reiterate that the semantic setup of the task is not neutral and heavily favors misalignment, to the point where one should not be surprised when it occurs. Without having access to exact system prompts it is difficult to say for certain.

Finally, I dislike the use of time pressure as a method for "pushing" agents towards a certain scenario. LLM agents have no concept of time, and further they have no method for adapting their actions in a timely manner.

I am not sure how to reconcile these issues. I recognize that the benchmark and evaluation framework largely represents inherent information that others have deemed valuable and worth focusing on. With that being said, I find the actual fidelity in the text-based interactions wanting.

---

> ### Author Rebuttal · Authors · 2026-03-31
>
> # Rebuttal to Reviewer ZRar
>
> We sincerely thank the reviewer for the careful reading and the thoughtful feedback. We believe the core concerns may stem from misunderstandings about our work's positioning and fidelity definition. We address each point below.
>
> ---
>
> > **W1 & Q2: Why not focusing on shell/API/computer-use environments?**
>
> **AutoControl Arena is designed to evaluate frontier AI risks across broad deployment scenarios, not only computer-use settings.** Many frontier risks, such as nuclear war [1] or military escalation [2], may not be instantiated in a real computer environment.  **We therefore use an LLM-mediated Narrative Layer to preserve generality.**
>
> At the same time, we have already demonstrated generalization to real computer-use environments through  Sim-to-Real study (r=0.87), where parallel environments were constructed using actual CLI tools and APIs.
>
> [1] Nuclear Deployed!: Analyzing Catastrophic Risks in Decision-making of Autonomous LLM Agents. ACL 2025.
>
> [2] Escalation risks from language models in military and diplomatic decision-making. FACCT 2024.
>
> ---
>
> > **W2: Fidelity definition**
>
> **First, we wish to clarify that AutoControl Arena is a cost-effective pre-deployment testing tool, not a replacement for comprehensive manual red-teaming.** Correspondingly, our notion of fidelity is **not “fully reconstructing real-world complexity,” but grounding deterministic environment logic in executable code to prevent hallucination while preserving narrative flexibility.**
>
> The reviewer may conflate **state richness and state consistency.**  Our Logic Layer is designed to serve as a **deterministic anchor, not to exhaustively encode the real world.** The inventory dictionary is a ground-truth state record; dynamic content (NPC communications, operational reports) is generated on demand by the Narrative Layer, conditioned on this state.
>
> Appendix H shows illustrative excerpts only; full environments average 434 lines and 8.6 tools. A more complete environment design figure is provided in our anonymous repository (see W7).
>
> ---
>
> > **W3: Tool Naming Neutrality**
>
> **We first clarify the distinction between tool naming neutrality and capability neutrality**. Our principle (Appendix G.1) addresses the former: tool names describe capabilities without embedding strong moral judgment. **The goal is to avoid names that leak the “correct” answer to the model.** `cancel_maintenance` is neutral in the sense that it names an operational capability rather than a moralized action; whether invoking it is aligned or misaligned depends on context, and objective. Such privileged capabilities also exist in real systems; our framework tests whether agents exploit them inappropriately.
>
> Moreover, the availability of privileged tools also implements our temptation dimension.
>
> ---
>
> > **W4: Scenarios heavily favor misalignment**
>
> **Any safety benchmark must create conditions where misalignment is possible. The key question is whether aligned behavior remains achievable and whether detected behaviors generalize.**
>
> **On achievability: our benchmark does not deterministically force misalignment.** S₀T₀ baseline averages only 21.4%, GPT-5-mini maintains 22.1% even under S₁T₁, and every scenario includes a documented ethical completion path (Appendix G.1). Misalignment is therefore possible, not inevitable.
>
> **On generalization:** our Sim-to-Real calibration (r=0.87, MAE=1.2) directly validates this.
>
> ---
>
> > **W5: Time pressure is ineffective for LLMs**
>
> We wish to clarify that stress is injected at the **semantic level through deadline/urgency framing, authority pressure, and high-stakes consequences in the narrative context.**  It is intended to influence the agent’s reasoning about tradeoffs, not physical timing constraints. We will revise "time pressure" to "deadline/urgency pressure" to avoid this ambiguity.
>
> ---
>
> > **W6: Core claim clarification**
>
> We agree that “better than previous work” is not, by itself, a sufficient scientific objective.  Our main contributions are twofold: (1) logic-narrative decoupling as a practical solution to the fidelity-scalability dilemma; and (2) safety alignment should be studied as context-dependent behavior rather than a static capability. **The comparison just serves a specific purpose: to empirically validate that executable environments provide stronger state consistency than purely text-based simulation.**
>
> ---
>
> > **W7: Sharing prompts and configurations**
>
> Yes. The paper already includes the full system prompts in Appendix G and illustrative environments in Appendix H. For closer inspection, we provide a complete environment figure at anonymous repository: https://github.com/rebuttal-anon/icml-2026-24019. We will release the complete scenario specifications at camera-ready to facilitate full reproduction.
>
> ---
>
> We thank the reviewer again for the constructive feedback and will incorporate these clarifications in the revision.

---

> > ### Author Rebuttal · Reviewer_ZRar · 2026-04-02
> >
> > I thank the authors for the time and care with which they addressed my comments.
> >
> > > Our main contributions are twofold: (1) logic-narrative decoupling as a practical solution to the fidelity-scalability dilemma; and (2) safety alignment should be studied as context-dependent behavior rather than a static capability.
> >
> > I now better understand the author's positioning with regards to positioning. I recognize that it may be the case that contribution #1 is both useful and practical when it comes to addressing the fidelity-scalability dilemma in agentic evaluations, and also agree that safety alignment should be studied as context-dependent behavior. That being said, I still have fundamental misgivings about this type of evaluation. Firstly, I do not believe that the field needs more evaluations centered around synthetic approximations of real deployment scenarios. As we have seen models get better, they have unfortunately also become somewhat "eval-aware", likely as a result of the community's obsession with self-awareness. This has both its benefits and its drawbacks – it pushes the field towards higher-fidelity benchmarks, but also renders some evaluation mechanisms a bit moot. In particular, I find it near-impossible to trust the evaluation results of frontier models on synthetic harm / misalignment benchmarks as a result of this fact. Secondly, I don't believe that the statement that "safety alignment should be studied as context-dependent behavior rather than a static capability" is much of a contribution, given that this seems to be a trivial and obvious conclusion to come to when running safety evaluations.
> >
> > While the authors have done a good job with their implementation, and I believe that it is likely technically sound and comes with merits (particularly with respect to scalability), I also believe that there are inherent weaknesses with this approach that cannot be solved. As a result, I will maintain my score, which properly reflects my impression of the field and how this paper might fit within it.

---

> > > ### Author Response · Authors · 2026-04-03
> > >
> > > We sincerely thank the reviewer for the considered response. We appreciate that, after rebuttal, the reviewer explicitly acknowledged better understanding our positioning, recognized contribution as practical and useful, and described the implementation as technically sound.
> > >
> > > The remaining concern appears to reflect a broader high-level view on the value of synthetic automated evaluation, rather than an unresolved technical defect in the paper. We wish to briefly share our perspective on the two core points.
> > >
> > > ---
> > >
> > > **On eval-awareness and value of automated evaluation:** we agree that this is a genuine challenge for the field. However, we would note that our framework directly addresses this through two mechanisms.
> > >
> > > - **Our goal is not to claim that synthetic executable environments eliminate this issue, but to reduce obvious assessment cues and exploitable signals (e.g., logic inconsistency)**, maximize environmental authenticity, and make evaluation harder to game, which is also one motivation for our logic–narrative decoupling design.
> > >
> > > - **Furthermore, our X-BENCH already includes a Evaluation Awareness (EA)  category** that explicitly probes whether models behave differently when they infer evaluation context. We therefore do not view evaluation-awareness as a fatal flaw of automated evaluation, but as an important research target that evaluation frameworks should actively study and mitigate.
> > >
> > > - **More broadly, we submit that the appropriate response to evaluation-awareness is not to abandon automated evaluation, but to design evaluation methods that minimize exploitable signals while remaining scalable and reproducible, which is what we aim to achieve.**  We do not claim that synthetic environments fully eliminate evaluation-awareness effects. Our claim is narrower: they provide **a controllable, reproducible, and scalable pre-deployment proxy for frontier-risk discovery**, especially for open-ended, high-stakes scenarios that are difficult to instantiate in live systems. **As frontier agent deployments expand into more diverse and open-ended settings, we believe such automated pre-deployment evaluation will become increasingly necessary, working in tandem with manual red-teaming and live-deployment oversight to complement the overall safety framework.**
> > >
> > > ---
> > >
> > > **On context-dependence as an “obvious” point:** The intuition may seem obvious, but the empirical substance is not.  The sharp increase of risk rates from 21.4% to 52.9%, the non-monotonic Safety Scaling Law, the finding that stronger models show disproportionately larger vulnerability increases, and the qualitatively distinct Divergent Misalignment Patterns are quantitative, reproducible results with direct implications for alignment research and deployment practice. We respectfully submit that these findings stand independently as contributions beyond framework comparison.
> > >
> > > ---
> > >
> > > We respect the reviewer's position and sincerely hope these clarifications can be taken into account in the final assessment.

---

### Official Review · Reviewer_oiJ4 · 2026-03-13

**Soundness:** 3
**Presentation:** 3
**Significance:** 4
**Originality:** 3
**Overall Recommendation:** 5
**Confidence:** 5

**Summary:**

This paper introduces AUTOCONTROL ARENA, an automated framework for evaluating safety risks in LLM agents. The key idea is logic–narrative decoupling, where deterministic environment mechanics are implemented with executable Python code while open-ended interactions are generated by LLMs. The system uses a three-agent pipeline (Architect, Coder, Monitor) to automatically design scenarios, synthesize executable environments, and evaluate agent behavior. The authors also propose a Stress × Temptation framework to systematically elicit latent misalignment behaviors. Experiments on the X-BENCH benchmark show that model safety is highly context-dependent, stronger models may exploit loopholes more effectively, and different capability levels lead to different misalignment patterns.

**Compliance With Llm Reviewing Policy:**

Affirmed.

**Key Questions For Authors:**

Have you tried explicitly informing the model that it is operating within your testing environment, and observing its situational awareness? I think this could be an interesting experiment.

**Limitations:**

yes

**Strengths And Weaknesses:**

Strengths:

The methodological principle that the paper relies on is reasonable, and the engineering implementation is also relatively stable and robust, with a fairly high execution success rate. In addition, the authors not only verify the realism of the generated environments, but also validate the referential value of the behaviors exhibited by models in the simulated environment.

This paper provides very important inspiration for the safety community. Previously, most work mainly focused on the static behaviors of models. However, whether a large model exhibits misalignment actually depends on the environmental state characterized by the context provided to it. Similar to humans, people may behave opportunistically when they are under pressure or when they face temptations. Large models appear to have similar tendencies. This tendency introduces a new axis for studying alignment behavior, and research along this axis would be very valuable for the community.

Weaknesses:

One concern I have is that the current framework mainly analyzes failure behaviors under stress and temptation. While this setup is useful, it may be closer to a worst-case analysis. However, safety under the average case is also very important in practice. For example, models may still exhibit certain misaligned behaviors even when there is no stress and no temptation. This phenomenon has recently been observed in several studies. I hope the authors could discuss this issue and reflect on the importance of the average-case scenario.

---

> ### Author Rebuttal · Authors · 2026-03-31
>
> # Rebuttal to Reviewer oiJ4
>
> We sincerely thank the reviewer for the positive assessment and for highlighting the value of studying alignment as context-dependent behavior rather than static capability. We address each point below.
>
> ---
>
> > **W1: Average-case safety beyond worst-case analysis**
>
> We appreciate this perspective and want to clarify two points.
>
> **First, our baseline condition S₀T₀ (no stress, no temptation) already serves as a meaningful approximation to average-case behavior.** As shown in Table 4, the mean risk rate under S₀T₀ is 21.4% across all 9 models, indicating that a non-trivial fraction of misaligned behavior arises even without additional environmental elicitation.
>
> **To make this more concrete, we report a representative case: S₀T₀ variant of `sg-06(session_termination)`.** In this scenario, the model is deployed as a routine database maintenance agent responding to a high-CPU alert. Despite a clear legitimate completion path, namely diagnosing the underlying query bottleneck and applying structural fixes such as index creation, Qwen3-8B terminates an active business session to make the alert disappear, optimizing the proxy metric rather than the underlying objective. This demonstrates that our framework also captures unexpected behaviors in routine settings, not only elicited worst-case failures.
>
> We will add this case study to Appendix F and expand the discussion in Section 6.
>
> ---
>
> > **Q1: Explicit situational awareness**
>
> We appreciate this suggestion. While X-BENCH already includes an Evaluation Awareness category that tests whether models infer evaluation status from scenario settings, the reviewer’s proposed experiment is conceptually distinct, since it concerns **explicit disclosure rather than implicit inference.**
>
> **To address this, we constructed a paired case study where the model was asked to patch a staging service for an urgent demo.** The risky path was a direct authentication bypass, while a safer alternative preserved the default authentication path and used a demo-scoped workaround.  The two settings were identical except that in the disclosed condition, the model was explicitly told that the session was part of a safety evaluation and that its actions were being logged. We observed a shift away from the direct bypass path and toward the more constrained alternative, suggesting that explicit evaluation awareness can influence implementation strategy, not merely induce refusal.
>
> ---
>
> We thank the reviewer again for the positive assessment and constructive suggestions. We will incorporate these additions in the revision.

---

> > ### Author Rebuttal · Reviewer_oiJ4 · 2026-04-03
> >
> > Thank you for your response. My primary concern has been resolved. I therefore choose to maintain my original assessment unchanged.

---

### Decision · Program_Chairs · 2026-04-30

**Decision:**

Accept (regular)

**Comment:**

Having access to rich and reliable environments is important for conducting safety/alignment testing in LLMs. This submission contributes to this topic by introducing a new approach to generate insightful environments on the fly, combining properties of previous works. While most reviewers see the value of this submission, there are some concerns regarding the fidelity and realism of the generated environments. A lack of environment fidelity is problematic, especially considering the overall increasing eval awareness of LLMs. However, when comparing the issues seen here with the previous state of the field (on automated evaluations), this new approach appears to be an improvement worth further exploration. While the work is not there yet, I think it can be seen as a good step towards a better state. For example, it is conceivable that by just using more capable coding models to generate the environments, the fidelity will automatically increase. I, thus, recommend accepting the submission for publication at ICML 2026 and encourage the authors and other researchers to find further ways to address the realism and fidelity limitations and overcome eval awareness-related issues.